# A Long Non-Coding RNA of *Citrus tristeza virus*: Role in the Virus Interplay with the Host Immunity

**DOI:** 10.3390/v11050436

**Published:** 2019-05-14

**Authors:** Sung-Hwan Kang, Yong-Duo Sun, Osama O. Atallah, Jose Carlos Huguet-Tapia, Jerald D. Noble, Svetlana Y. Folimonova

**Affiliations:** 1Department of Plant Pathology, University of Florida, Gainesville, FL 32611, USA; shkang@ufl.edu (S.-H.K.); yd.sun@ufl.edu (Y.-D.S.); oosman@zu.edu.eg (O.O.A.); jhuguet@ufl.edu (J.C.H.-T.); 2Plant Molecular and Cellular Biology Program, University of Florida, Gainesville, FL 32611, USA; jnoble333@ufl.edu

**Keywords:** RNA virus, closterovirus, Citrus tristeza virus, long non-coding RNA, plant immunity, salicylic acid signaling, alternative oxidase, pathogenesis-related genes

## Abstract

During infection, *Citrus tristeza virus* (CTV) produces a non-coding subgenomic RNA referred to as low-molecular-weight tristeza 1 (LMT1), which for a long time has been considered as a by-product of the complex CTV replication machinery. In this study, we investigated the role of LMT1 in the virus infection cycle using a CTV variant that does not produce LMT1 (CTV-LMT1d). We showed that lack of LMT1 did not halt virus ability to replicate or form proper virions. However, the mutant virus demonstrated significantly reduced invasiveness and systemic spread in *Nicotiana benthamiana* as well as an inability to establish infection in citrus. Introduction of CTV-LMT1d into the herbaceous host resulted in elevation of the levels of salicylic acid (SA) and SA-responsive pathogenesis-related genes beyond those upon inoculation with wild-type (WT) virus (CTV-WT). Further analysis showed that the LMT1 RNA produced by CTV-WT or via ectopic expression in the *N. benthamiana* leaves suppressed SA accumulation and up-regulated an alternative oxidase gene, which appeared to mitigate the accumulation of reactive oxygen species. To the best of our knowledge, this is the first report of a plant viral long non-coding RNA being involved in counter-acting host response by subverting the SA-mediated plant defense.

## 1. Introduction

During host infection, viruses produce a vast number of different RNA species. Besides generating mRNAs for translation of their proteins, many viruses make non-coding RNAs (ncRNAs) [1,2,3]. While functions of viral small ncRNAs such as small interfering RNAs (siRNAs) and microRNAs are relatively well understood, the roles of long ncRNAs (lncRNAs; RNAs over 150 nucleotides, nt) remain understudied. There also are fewer viruses for which lncRNAs have been described or investigated at the mechanistic level. Yet, a number of those lncRNAs have been assigned various functions in the virus cycle favoring infection, including regulation of host and viral gene expression, virion production, and counteraction of the host antiviral response. Some of the best studied examples are lncRNAs produced by large double-stranded DNA viruses in the *Adeno-* and *Herpesviridae* families and non-coding subgenomic RNAs (sgRNAs) that are generated by positive-sense RNA viruses of the family *Flaviviridae* (reviewed in [2]). In addition to the lncRNAs of animal viruses, few lncRNAs were found in plant-infecting positive-sense RNA viruses belonging to the Luteo- and Tombusviridae families and in *Cauliflower mosaic virus* (CaMV), a pararetrovirus from the family Caulimoviridae with a nicked, double-stranded circular DNA genome (reviewed in [3]). Despite being produced at different stages of the viral cycle via different mechanisms, lncRNAs of different viruses share many commonalities. All these lncRNAs are produced in high molar excess. Typically, they are not required for virus replication. However, they increase virus pathogenicity greatly and carry out a notable role in the evasion of the host immunity. Often, this is achieved through the association of viral lncRNAs with the host proteins that inhibits the activity of the latter proteins or diverts them from their normal cellular processes (reviewed in [2,3]).

In plants, two major mechanisms mediate antiviral defense. The first is RNA silencing, which is triggered by viral double-stranded RNA that are processed into siRNAs of 21–24 nt in size by the host RNA silencing machinery, which in their turn guide degradation of RNA sequences in a homology-dependent manner [4,5,6]. To overcome such defense response, viruses express protein suppressors of RNA silencing that interfere with various steps of this process [7]. A few of them were also shown to produce ncRNAs, which generate massive amounts of viral siRNAs that serve as sponges for the host components of the RNA silencing machinery and, thus, divert them from degrading other viral RNA species. One such example is CaMV 8S RNA [3,8,9].

The second mechanism involves phytohormone (e.g., salicylic acid (SA), 2-hydroxybenzoic acid)-mediated signaling pathways. SA is a multi-faceted hormone, which plays a role in plant development and response to biotic stress (reviewed in [10,11,12,13]). With viral pathogens, SA was shown to be a key compound in the signal transduction triggering plant defense responses resulting in suppression of virus amplification and movement ([14,15,16], reviewed in [17]). Induction of SA upon pathogen perception leads to activation of defense-related genes, including those encoding several families of pathogenesis-related (PR) proteins ([18,19], reviewed in [20]). Some viruses were found to produce proteins that interfere with SA signaling [21,22]. However, to date, no studies reported an involvement of virus-generated lncRNAs in mitigation of SA-regulated defense pathways.

*Citrus tristeza virus* (CTV), one of the largest plant RNA viruses, which belongs to the family Closteroviridae, produces a plethora of different RNA species in the infected cells. Those include a positive-strand genomic RNA and its complementary negative-strand copy, more than 30 3′- and 5′-terminal sgRNAs of both polarities, a number of defective RNAs, which sequence composition, size, and amount greatly vary depending on the virus strain and virus-propagating host, as well as a large number of viral siRNAs that are generated by the host RNA silencing machinery in response to virus invasion [23,24,25,26,27]. The 19.3 kb genomic RNA of CTV contains 12 open reading frames (ORFs) and serves as mRNA for translation of ORFs 1a and 1b, which encode proteins required for virus replication (Figure 1A) [23]. The other ten ORFs are translated via 3′ co-terminal sgRNAs which production is driven by the respective controller elements (CEs) functioning as sgRNA promoters and/or terminators [23,24,25,28]. These sgRNAs direct synthesis of the major (CP) and minor (CPm) coat proteins, p65 (a homologue of cellular HSP70 proteins), and p61 that are required for virion assembly and movement along with the p6 protein [29,30]; p20 and p23 proteins known to be viral suppressors of RNA silencing along with CP [31]; and p33, p18, and p13 proteins, which play a role in extending the virus host range [32,33]. Each 3′ co-terminal sgRNA is accompanied by a complementary negative-strand sgRNA and a 5′-terminal positive-strand sgRNA, all of which arise due to promoting and/or terminating activity of the same CE [24,25]. The positive- and negative-strand 3′ co-terminal sgRNAs are generated in large amounts, whereas the 5′-terminal sgRNAs are made in the minor quantities and are thought to result from termination of the genomic RNA synthesis at the corresponding CEs as by-products of the CTV replication machinery.

In addition to the above sgRNA species, two unusual small (~ 750 nt and ~ 650 nt) 5′ co-terminal positive-strand sgRNAs referred to as low-molecular-weight tristeza 1 and 2 (LMT1 and LMT2, respectively) are abundantly produced during CTV infection [25,34,35,36]. In contrast to the 3′ co-terminal positive-strand protein-coding sgRNAs, LMT 1 and 2 do not have corresponding negative-strand sgRNAs. Both LMTs were shown to be generated by all tested strains of CTV [35,36]. Interestingly, a similarly-sized 5′-terminal positive-strand sgRNA has been also observed for another closterovirus, *Beet yellows virus* [37]. Although the LMT 1 and LMT2 sgRNAs were reported more than two decades ago [34], their functions, in particular that of LMT1, remain obscure. The mechanism of the LMT1 production has been resolved: LMT1 is generated during replication by initiation at the 3′end of the negative-strand copy of the genomic RNA and termination at the CE located in the structural region containing two stem loops mapped to the nt 719–775 in the CTV genome of the T36 strain [25]. Importantly, LMT1 is made early in the infection [25]. On the other hand, no CE activity was detected for LMT2. This RNA was shown to arise later during virion assembly process and represent a part of the viral RNA encapsidated by CPm [36]. However, the precise mechanism and functional importance of the LMT2 accumulation have not been found. Furthermore, no specific function has been identified for LMT1 either. The LMT1 sgRNA does not have an ORF and appears to be a lncRNA of CTV. Prevention of its production by introducing mutations within the corresponding CE region was shown not to impede virus replication in protoplasts, and it has been suggested that the formation of LMT1 could be accidental [25].

In this study, we examined a role of LMT1 in CTV infection using a virus mutant that does not produce LMT1 (CTV-LMT1d). Lack of the LMT1 production did not halt virus ability to amplify in the initially inoculated cells of *Nicotiana benthamiana* or form proper virions. Yet, the level of the CTV-LMT1d accumulation was reduced compared to that of the parental wild-type (WT) CTV. Moreover, the inability to produce LMT1 resulted in a significant decrease in the virus invasiveness and systemic spread in this herbaceous host. The observed phenotypic differences correlated with a notable difference in the SA-mediated host immune responses to the wild type and the LMT1-deficient viruses, implicating the LMT1 RNA in counter-acting host defenses. Furthermore, LMT1 was found indispensable for the CTV infection of the natural citrus host. Taken together, these results suggest that CTV has evolved the lncRNA LMT1 to evade host immune system in order to facilitate virus infection. To the best of our knowledge, this is the first report for a plant viral lncRNA being involved in subverting the SA-mediated plant defense.

## 2. Materials and Methods

### 2.1. Generation of Constructs

#### 2.1.1. pCTV-LMT1d

pCTV-LMT1d was generated using an infectious cDNA clone of the T36 isolate of CTV tagged with the green fluorescent protein (GFP) ORF placed under the CaMV 35S promoter in the binary vector pCAMBIA1380 (pCTV-GFP; [38]). A fragment encompassing the region in the CTV genome between *Asc*I to *Bsu*36I restriction endonuclease recognition sites was synthesized with the alteration of six nucleotides (nt 722, 725, 728, 731, 734, and 737) using custom gene synthesis service and cloned into the pUC58 vector (GenScript, Piscataway, NJ, USA). The synthesized fragment in the pUC58 vector was digested using *Asc*I and *Bsu*36I restriction endonucleases and ligated into the corresponding region of pCTV-GFP, which was digested with the same restriction endonucleases. The virus constructs obtained were screened by restriction endonuclease (*EcoR*V) digestion pattern and confirmed by sequencing. The engineered CTV variants were used for inoculation of *N. benthamiana* and *Citrus macrophylla* plants as described below. Conservation of the parental sequence including introduced mutations in the virus progeny was verified by sequence analysis of the reverse transcription polymerase chain reaction (RT-PCR) products obtained using total RNA extracts from virus-inoculated plants and primers corresponding to the modified region. Sequence analysis was performed at Macrogen Inc. (Rockville, MD, USA).

#### 2.1.2. pLMT1 and Its Mutants

For the construction of pLMT1, a fragment of the LMT1 region placed under the CaMV 35S promoter sequence was amplified using pCTV-GFP as a template with a pair of primers annealing to the 35S promoter and nt position 800 in the CTV genome (35S-F; 5′-AAGGGTAATATCCGGAAACCTCCTCGGATTCCATTGCCCAGCT-3′ and LMT1-R; 5′-TTCGAGCTGGTCACCTGTAATTCACACGTGTCGAGGGATGAGGATTAACCTCTTCGA-3′, respectively) and subsequently substituted using an In-Fusion HD Cloning Kit (Takara Bio, Mountain View, CA, USA) for the corresponding fragment excised from the linearized pCAMBIA vector by digestion using restriction endonucleases *BspE*I and *Pml*I. pLMT1M1 and pLMT1M2 were constructed similarly. The substitution fragment of LMT1M1 carrying an alteration of thymine (T) to cytosine (C) at nt 109 was amplified by overlap-extension polymerase chain reaction (OE-PCR) using two pairs of primers 35S-F and OE-R1 (5′-AGTGTAATTTTTGTTGTGGGAATATTA-3′) for fragment 1 or OE-F2 (5′-TATTCCCACAACAAAAATTACACTACGTCGAAACTC-3′) and LMT1-R for fragment 2. Two overlapping fragments were used as templates to amplify the final product using a pair of primers (35S-F and LMT1-R). The substitution fragment of LMT1M2 containing an alteration of Ts to Cs at nt 109 and 265 was synthesized through custom gene synthesis service and cloned into the pUC58 vector (GenScript, NJ, USA). The synthesized fragment in the pUC58 vector was used as a template to amplify a fragment to substitute the corresponding region of pLMT1. The oligonucleotide primers 35S-F and LMT1-R were used to amplify the fragment. Both LMT1M1 and LMT1M2 fragments were subsequently substituted in pCAMBIA to generate the final product as described above for pLMT1.

### 2.2. Agroinfiltration of CTV Constructs into N. benthamiana

Agroinfiltration of CTV constructs was conducted as previously described [39]. Briefly, *Agrobacterium tumefaciens* EHA105 was transformed by heat shock with the cloned plasmids, and the resulting transformants were selected on the Luria–Bertani agar plates containing two antibiotics (50 µg/mL rifampicin and 25 µg/mL kanamycin). Over-night cultures of a selected single colony were gently resuspended in a buffer containing 10 mM 2-(*N*-morpholino) ethane sulfonic acid (MES, pH 5.85), 10 mM MgCl_2_ and 150 mM acetosyringone at the optical density (OD)_600nm_ = 0.1 or 1.0. Following three-hour incubation at room temperature without shaking the suspension was infiltrated into three-week-old *N. benthamiana* plants using needless syringe.

### 2.3. Inoculation of Citrus Plants

The infiltrated leaves of *N. benthamiana* plants tested positive for CTV infection by RT-PCR and fluorescence microscopy were harvested at seven dpi and used for virion extraction as previously described [40]. The extracted virions were used for “bark-flap” inoculation of nine- to 12-month old *C. macrophylla* trees. The concentration of the prepared virions was adjusted based on the enzyme-linked immunosorbent assay (ELISA) to the absorbance value (*A*)_405 nm_ = 2.5 to ensure the introduction of an equal amount of inoculum. The inoculated citrus plants were maintained under the greenhouse conditions until further analysis.

### 2.4. Examination of Fluorescence in Plants

GFP fluorescence in the infiltrated leaves of *N. benthamiana* plants was observed using a hand-held ultraviolet (UV) lamp (365 nm, UVP, USA) set 6 inches above the plants in a dark room. Leaves of *N. benthamiana* plants and bark tissue from citrus trees were examined for GFP fluorescence using a Leica MZ10F fluorescence dissecting stereomicroscope with an attached Leica DFC310FX camera (Leica Microsystems, Wetzlar, Germany) and processed using Leica LAS-X (Leica Microsystems).

### 2.5. Total RNA Extraction

150 mg of bark tissue collected from young flushes of *C. macrophylla* or leaf discs (1 cm in diameter) from *N. benthamiana* was ground in liquid nitrogen, and total RNA was extracted using the Trizol Reagent (Thermo Fisher Scientific, Waltham, MA, USA) according to the procedure of the manufacturer. RNA extracts were resuspended in 100 µL of RNase-free water. RNA concentration was measured using a NanoDrop Ultraviolet–Visible (UV–Vis) Spectrophotometer (Thermo Fisher Scientific, Waltham, MA, USA). The RNA extracts were diluted with RNase-free water to adjust concentration to 100, 10 and 1 ng/µL. Aliquots were stored at −80 °C until use.

### 2.6. Northern Blotting

Total RNA (0.5 µg) was separated by electrophoresis in formaldehyde denaturing 1% agarose gels in MOPS buffer (20 mM MOPS, 5 mM sodium acetate, 1 mM EDTA, pH 7.0) and transferred to Biodyne™ B nylon membranes (Thermo Fisher Scientific, Waltham, MA, USA) using Trans-blot SD semi-dry transfer system (Bio-Rad, Hercules, CA, USA). Hybridization of digoxygenin (DIG)-labeled riboprobes specific to the positive-strand RNA corresponding to the 5′ or 3′ end of the CTV genome generated as described in Satyanarayana et al. [41] was carried out with ULTRAhyb™ Ultrasensitive hybridization buffer (Thermo Fisher Scientific, Waltham, MA, USA) and detection was followed as described in Mawassi et al. [34] and DIG-Northern-system manufacturer’s instructions (Roche, Pleasanton, CA, USA).

### 2.7. Viral Load Quantification

Viral load in the inoculated plants was analyzed by RT-quantitative (q)PCR using total RNA as a template and determined using a standard curve as previously described [42]. Briefly, RT-qPCR was performed in a CFX Connect^TM^ Real-time PCR detection system (BIO-RAD) using the SuperScript^®^ III One-Step RT-PCR kit (Invitrogen, USA) with 2 µL of total RNA extract (1 ng/µL), 5 µL of 2× reaction mix, 0.4 µM of the forward (T36-RT-F; 5′-ACTTCGGACAAGCGGGTGAATT-3′) and reverse primers (T36-RT-R; 5′-GCAAACATCTCGACTCAACTACC-3′) corresponding to the intergenic region between the RNA-dependent RNA polymerase and p33 ORFs of CTV and 0.1 µM of 6-FAM/BHQ-1 labeled Taqman probe (5′-AGCAACCGGCTGATCGATTGATT-3′) in a total reaction volume of 10 µL. Cycling conditions included reverse transcription at 50 °C for 5 min, incubation at 94 °C for 2 min, and 40 cycles of 94 °C for 10 s and 60 °C for 40 s. Control samples in each run included total RNA from healthy citrus, water control, and at least two RNA transcript dilutions generated for the standard curve.

### 2.8. Enzyme-Linked Immunosorbent Assay (ELISA)

0.25 g of bark tissue from young flushes of *C. macrophylla* or leaf tissue from *N. benthamiana* plants ground in 5 mL of the extraction buffer (3.2 mM Na_2_HPO_4_, 0.5 mM KH_2_PO_4_, 1.3 m KCL, 135 mM NaCl, 0.05% Tween^®^ 20, pH 7.4) per sample was used for double-antibody sandwich ELISA as per manufacturer’s instructions (Agdia, Elkhart, IN, USA) to confirm presence of viruses.

### 2.9. Virion Analysis

10 μL-drops of virion extracts were placed on the 400 mesh copper grids (Formvar/Carbon Square Mesh, UB, Electron Microscopy Sciences, Hatfield, PA, USA) for negative contrast electron microscopy. After 5 min incubation, the virion solution on each grid was removed using filter paper. The grids were stained using 1% uranyl acetate dissolved in distilled water for 30 s followed by drying in the air. Images were taken using Hitachi H-7000 transmission electron microscope (TEM, Hitachi High-Technologies, Schaumburg, IL, USA) equipped with a Veleta (2k × 2k) CCD side mount camera (EMSIS, Munster, Germany) at 100 kV. The length of virions was measured using Image J software (http://rsbweb.nih.gov/ij) directly from the images taken. For each mutant virus, over 100 flexuous virions were measured and up to 10 longest virions were selected to calculate the mean value.

### 2.10. Ribonuclease Digestion

Virions prepared as described above were diluted (1 mg/200 μL) in nuclease digestion buffer (10 mM Tris-HCl pH 8.0, 1 mM CaCl_2_). Upon the addition of 5 units of micrococcal nuclease (New England Biolabs, Ipswich, MA, USA), digestion was carried out by incubation in 37 °C water bath for an hour. A “stop solution” (8.6% SDS, 0.007 M EDTA) was added to halt the reaction.

### 2.11. Relative Expression Levels of Salicylic Acid (SA)-Responsive Genes

The relative expression levels of SA-responsive genes were evaluated by RT-qPCR usig Luna^®^ universal RT-qPCR system (New England Biolabs, Ipswich, MA, USA) as per the manufacturer’s instructions. Briefly, 2 µL of total RNA (10 ng/µL) extracted as described above with a pair of 0.5 µM oligomers for *PR-1a/c*, *PR-2* and *PR-5* [43] and *NbAOX1* [44] in total 20 µL-reaction mixtures were reverse-transcribed and amplified using a CFX Connect^TM^ Real-time PCR detection system (BIO-RAD). A pair of oligomers amplifying ubiquitin (UBQ) gene (NtUBQ-F; 5′-TCCAGGACAAGGAGGGTATCC-3′ and NtUBQ-R; 5′- TAGTCAGCCAAGGTCCTTCCA-3′) was used as an internal control. The cycling conditions included reverse transcription at 55 °C for 10 min, incubation at 95 °C for 2 min, and 40 cycles of 95 °C for 10 s and 60 °C for 60 s. Ct values were averaged for triplicates of each sample from three independent RT-qPCR reactions prior to calculating relative values using the 2^−ΔΔCt^ method [45].

### 2.12. Total SA Measurement

Quantification of SA content in the plants was determined as previously described [46]. Briefly, a set of 500 mg tissue homogenized using 1600 MiniG^®^ homogenizer and resuspended in 250 µL of 0.1 M acetate buffer was incubated in the presence (for the glucose-conjugated SA, SAG) or absence (for free SA) of β-glucosidase at 37 °C for 90 min. The incubated extract was distributed into three wells (20 µL/well) and mixed with 60 µL LB and 50 µL *Acinetobacter* sp. ADPWH_*lux* culture (OD_600nm_ = 0.4) followed by incubation at 37 °C for 60 min. The luminescence was read using a Synergy HTX plate reader (BioTek, Winooski, VT, USA).

### 2.13. Inoculations with Pseudomonas syringae

A pure culture of *P. syringae* (isolated from tobacco plants in Florida) was resuspended in 10 mM MgCl_2_ and adjusted to 10^3^ colony-forming unit (CFU)/mL. *N. benthamiana* leaves pre-inoculated with *Agrobacterium* cultures harboring pCTV-WT, pCTV-LMT1d, pLMT1 or empty vector were sequentially inoculated with the bacterial suspension using a needless syringe at 4 or 7 days after initial infiltration with pLMT1/empty vector or pCTV-WT/pCTV-LMT1, respectively. Leaf discs were collected from infiltrated areas at 3 or 4 days after challenge infiltration, respectively, and placed into 500 µL of 10 mM MgCl_2_. Tissue samples were homogenized using 1600 MiniG^®^ high-throughput homogenizer (SPEX, Metuchen, NJ, USA) at 1500 strokes per minute for 2 minutes and serially diluted to 10^1^–10^5^ folds. 20 µL of serially diluted samples were dropped on King’s B agar (Thermo Fisher Scientific, Waltham, MA, USA) plate and incubated at 28 °C for 48 h, and colonies were counted. The number of CFU per collected leaf disc was determined through the formula: CFU/leaf disc = (dilution factor × colony count/20) × 500/8. Statistical analysis was done as described above.

### 2.14. Histochemical Assay

3,3′-Diaminobenzidine (DAB) staining was performed as previously described [47]. Briefly, *N. benthamiana* leaves were washed with double distilled H_2_O three times and stained with 1 mg/mL DAB-HCl solution (Sigma-Aldrich, St. Louis, MO, USA) overnight. The samples were de-stained in 100% ethanol. The DAB stain intensity was quantified by measuring pixel intensity of infiltrated area using Fiji software [48].

## 3. Results

### 3.1. Prevention of LMT1 Production Does Not Affect Virus Ability to Replicate or Assemble Proper Virions

To investigate the role of LMT1 in the virus infection cycle, we generated a full-length cDNA clone of a mutant CTV variant that does not produce LMT1, CTV-LMT1d, by introducing a six nt-substitution into the CE driving production of the LMT1 sgRNA during virus replication (Figure 1A). As shown in an earlier study, silent substitutions of the nt 722, 725, 728, 731, 734, and 737 in the virus genome, which alter the RNA sequence of the corresponding LMT1 CE without affecting translation of the ORFs 1a/1b, prevent generation of LMT1 [25]. The cDNA constructs of the wild-type CTV (CTV-WT) and CTV-LMT1d were modified to accommodate an extra GFP ORF as a reporter gene to monitor virus accumulation in the plants. The entire virus expression cassettes were placed under the CaMV 35S promoter in the binary vector plasmid, which allowed their introduction into *N. benthamiana* plants by agroinfiltration.

To assess whether the introduced mutations affected production of LMT1 only, without a detrimental effect on other viral sgRNAs, we examined sgRNAs generated by CTV-WT and CTV-LMT1d using total RNA from the *N. benthamiana* leaves agroinfiltrated with pCTV-WT or pCTV-LMT1d construct (Figure 1B,C). Northern blot analysis of RNA extracted from the inoculated leaves at seven days post-infiltration (dpi) demonstrated similar production of 3′ co-terminal sgRNAs from both CTV-WT and CTV-LMT1d (Figure 1B). On the other hand, RNA hybridization performed using the 5′ end-specific probe confirmed that the LMT1 sgRNA was generated only by the wild-type virus but not by the CTV-LMT1d mutant. With CTV-WT, the LMT1 sgRNA accumulation was seen as early as at 4 dpi, and its amount increased by 7 dpi. No corresponding band was detected in the samples from the CTV-LMT1d-expressing tissue (Figure 1C). As we expected, production of LMT2, which typically accumulates later in the infection, was not affected by the introduced mutations: the LMT2-specific band was detected using samples collected at 7 dpi from the tissue infected by CTV-WT as well as from that infected by CTV-LMT1d (Figure 1C). These results confirmed that the introduced mutations eliminated production of LMT1 without affecting other viral sgRNAs.

At the next step, we investigated whether prevention of LMT1 production had any effect on the assembly of virus particles. Closterovirus virions are built of two coat proteins: the CPm, which covers a 5′-terminal region of the genomic RNA (in CTV, this region is ~630 nt), and the CP that encapsidates the rest of the genome [30,49,50,51]. Additionally, it has been suggested that the p65 and p61 proteins participate in virion formation as well and associate with the CPm-CP transition zone [52]. The substituted nucleotides in CTV-LMT1d are positioned in the proximity of the genomic region corresponding to that transition area, and, thus, their alteration could have a negative effect on the encapsidation process. To assess virion formation by CTV-LMT1d, virions isolated from the *N. benthamiana* leaves infiltrated with the corresponding infectious clone were examined using transmission electron microscopy (Figure 1D). As we observed, the average length of virions isolated from the CTV-LMT1d-inoculated leaves was not significantly different from that of the CTV-WT virions (Figure 1E; Tukey’s HSD test post hoc, *p* values < 0.01), which suggested that mutations within the LMT1 CE did not affect virion assembly. To further confirm proper packaging of the CTV-LMT1d genome at the LMT1 CE region, purified virions were treated with a micrococcal nuclease (MNase) and examined by RT-PCR to verify whether the genome was protected from the MNase activity (Figure 1F). Using a pair of oligonucleotides designed to amplify the region of the genome that is located downstream of the LMT1 CE and encodes the L1 protease domain (Figure 1A, L1Pro) we were able to obtain the expected fragment in the samples prepared from the CTV-WT- or CTV-LMT1d-infiltrated leaves. This suggested that the corresponding regions of their genomes were protected from the degrading enzyme by being encapsidated within the virions (Figure 1F, bottom panel labeled as “L1Pro”). Furthermore, RT-PCR reactions with a pair of oligonucleotides amplifying the 5′-terminal genomic segment encompassing sequences upstream of the LMT1 CE also resulted in amplification of the respective products in the CTV-WT and CTV-LMT1d samples, confirming proper encapsidation of the 5′-terminal regions of the two viral genomes (Figure 1F, top panel labeled as “5′”). Taken together, these results confirmed that mutations introduced to generate CTV-LMT1d successfully disabled production of the LMT1 sgRNA without disturbing other fundamental virus properties related to its replication and assembly.

### 3.2. The LMT1-Deficient Virus Shows an Impediment in its Invasiveness

Next, we evaluated and compared the levels of the CTV-WT and CTV-LMT1d accumulation upon their introduction in the *N. benthamiana* leaves by *Agrobacterium*-mediated infiltration of the respective GFP-tagged constructs. For both virus variants, examination of the infiltrated leaves using a hand-held UV lamp revealed production of a strong GFP fluorescence indicating their successful multiplication. In either case, the fluorescence was present in numerous cells distributed throughout the leaf lamina areas located between the vascular bundles (Figure 2A,B). However, there was an obvious difference between the two treatments in the GFP production in the cells proximal to the leaf veins. With CTV-WT, many more cells adjacent to the major and minor veins were showing GFP fluorescence (Figure 2A’), which compared to GFP being omitted in such areas in the leaves infiltrated with CTV-LMT1d (Figure 2B’). Further observations of the cross-sections through the leaves infiltrated with pCTV-WT using higher magnification showed GFP expression in the cells extending into the vascular bundles (Figure 2C). By contrast, no GFP fluorescence was found in the corresponding areas of the leaves infiltrated with pCTV-LMT1d (Figure 2D).

The absolute quantification of the viral titers measured by RT-qPCR using total RNA extracted from the infiltrated leaves showed that the levels of the CTV-WT and CTV-LMT1d accumulation were comparable at the initial infection stages (up to 7 dpi; Figure 2E). However, as the infection progressed CTV-WT demonstrated a significant increase in its titer compared to that of CTV-LMT1d, with the titer of the former virus exceeding that of the mutant variant by approximately four times at 10 dpi (Figure 2E). These results suggested that LMT1 plays an important role during the establishment of infection and contributes to virus invasiveness.

To assess the ability of CTV-WT and CTV-LMT1d to develop systemic infection, the infiltrated *N. benthamiana* plants were monitored for the spread of GFP fluorescence to the upper leaves (Figure 2F). At 16 dpi, the appearance of GFP fluorescence in the upper non-infiltrated leaves was noticed in more than half of the plants inoculated with CTV-WT (53%) and only in a low proportion of plants (13%) inoculated with CTV-LMT1d (Figure 2G). The development of systemic infections in those plants and the lack of that in the others in which the upper leaves remained virus-free was confirmed by the RT-PCR analysis with the oligonucleotides amplifying the CTV CP coding region (data not shown). These results indicated that lack of LMT1 negatively affected the ability of the virus to spread throughout this herbaceous host.

Our next goal was to examine whether LMT1 had a similar effect on virus infection in the natural citrus host. Virion preparations from the *N. benthamiana* leaves infiltrated with pCTV-WT or pCTV-LMT1d resuspended at a similar virus concentration based on the ELISA measurements were used for bark-flap inoculation of *C. macrophylla*, one of the most CTV-susceptible citrus varieties (see Materials and Methods). At three months post-inoculation, strong GFP fluorescence was detected in the phloem cells of the newly emerged tissue in most citrus trees inoculated with CTV-WT, indicating the establishment of successful infection. In contrast, none of the trees inoculated with CTV-LMT1d showed signs of infection (Figure 2H). By 12 months after inoculation, ~90% of trees inoculated with CTV-WT developed systemic infection (28 CTV-positive trees out of total 32 trees). None of 106 plants inoculated with the CTV-LMT1d virions became infected. The establishment of CTV-WT infection and the lack of the virus in the CTV-LMT1d-inoculated citrus plants were confirmed by ELISA with the CTV-specific antibody (Figure 2I). The inability of the virus variant lacking LMT1 to infect citrus indicated that LMT1 is essential for virus infection of the natural host.

### 3.3. Lack of LMT1 Restricts Virus Cell-to-Cell Movement in N. benthamiana

As shown above, lack of the LMT1 production had a negative effect on virus ability to invade the plant hosts. The LMT1-deficient virus was unable to infect citrus. In the herbaceous host, introduction of the mutant virus resulted in lesser number of plants that became infected, compared to a large proportion of plants that were infected with CTV-WT around the same time point. These outcomes correlated with the characteristic distribution of CTV-LMT1d in the initially inoculated leaves of *N. benthaminana* in which the latter showed a decreased ability to enter and/or propagate in cells forming the vascular network. Based on these observations, we hypothesized that the impeded movement capabilities of the mutant virus might account for the differences in both the local and systemic invasiveness found between CTV-WT and CTV-LMT1d. In order to compare the cell-to-cell movement of the wild-type and LMT1-deficient virus variants, we took an advantage of a previous observation that while CTV remains restricted to the phloem in the infected citrus plants, in *N. benthamiana*, the virus is able to move cell-to-cell between different types of cells, including mesophyll cells, though, not efficiently [53]. Virion suspensions prepared from leaves infiltrated with the GFP-tagged CTV-WT or CTV-LMT1d of approximately equal concentration verified by ELISA were rub-inoculated onto *N. benthamiana* leaves (see Materials and Methods). Examination of leaves at 21 dpi using a fluorescence microscope revealed formation of GFP-expressing foci composed of few mesophyll cells in the leaves inoculated with the wild-type and the mutant viruses (Figure 3A,B, respectively). The overall efficiency of rub-inoculation was low, which is in agreement with previous observations of a limited susceptibility of this plant species to mechanical inoculation with closteroviruses [53,54]. However, there was a visible difference in the number of fluorescent foci produced by the two virus variants and their size. Specifically, in the five independent replicate experiments, analysis of 100 leaves inoculated with each virus variant showed 59 and 32 fluorescent foci produced by CTV-WT and CTV-LMT1d, respectively. Importantly, only 8% of foci generated by the wild-type virus were unicellular, while the rest of foci were multicellular, containing up to 12 cells per cluster (Figure 3A). In contrast, 60% of CTV-LMT1d foci were unicellular and the rest were small clusters composed of two to three cells (Figure 3B). The quantification of the area of the fluorescence measured by the pixels with fluorescence from the images of the infected foci showed that the fluorescence-expressing area produced by CTV-WT was nine times larger than that of CTV-LMT1d (Figure 3C). These results suggested that LMT1 plays a role in facilitating the virus cell-to-cell movement.

### 3.4. Infection by *Citrus Tristeza Virus*-LMT1d (CTV-LMT1d) is Accompanied by Higher Expression Levels of SA-Responsive Defense-Related Genes Compared to That upon the CTV-WT Infection

In order to establish a successful infection, a virus must counter-act host defense responses. In plants, SA-mediated signaling pathways can suppress all three main stages in virus infection: replication, cell-to-cell movement, and long-distance movement [17]. Furthermore, it has been shown that SA signaling is involved in plant defense against CTV [55]. Therefore, we next tested if the host immune responses to CTV-WT and CTV-LMT1d were different. Before assaying the SA levels, we first examined the mRNA levels of the three PR genes associated with the SA pathway—the *PR-1a/c*, *PR-2*, and *PR-5* genes—in the leaves of *N. benthamiana* plants infiltrated with CTV-WT or CTV-LMT1d (Figure 4A). RT-qPCR quantification using total RNA extracted from the samples collected at 1, 4, 7, and 10 dpi demonstrated that at all time points starting at 4 dpi the three PR genes were expressed at significantly higher levels in the plants infected with the LMT1-deficient virus compared to those in the plants infected by CTV-WT. Since the difference in the PR genes expression was significant, we next analyzed the total SA levels composed of free SA and SAG using tissue samples from the above plants (Figure 4B). Quantification of the SA accumulation showed a significant difference between CTV-WT and CTV-LMT1d, with the SA accumulation in the CTV-LMT1d-infected tissue being two to three times higher than that in the CTV-WT-infected tissue (Figure 4B).

The up-regulation of the PR transcripts upon infection with pathogens is often used as a marker of the induction of the host immune responses [56,57,58]. The accumulation of the host defense compounds in response to a primary pathogen infection contributes to the defense against subsequent attacks by other pathogens [17,59,60,61,62]. Thus, we next compared the effect of the expression of the two virus variants on the sequential infection by *Pseudomonas syringae*. *N. benthamiana* plants were infiltrated with the *Agrobacterium* cultures harboring pCTV-WT or pCTV-LMT1d followed by infiltration of *P. syringae* into the same leaves at seven dpi. At four days after the infiltration with *P. syringae*, the inoculated leaves demonstrated the development of typical water-soaking symptoms, the severity of which was greater in the leaves expressing the wild-type virus (Figure 4C). Leaf discs from the leaves shown in Figure 4C were collected and subjected to the CFU counting assay on the King’s B medium. The leaves expressing CTV-WT had a significantly higher CFU count of *P. syringae* than that in the leaves expressing CTV-LMT1d (Figure 4D). In other words, plants pre-infiltrated with pCTV-LMT1d were less susceptible to the secondary biotrophic infection, compared to those pre-infiltrated with pCTV-WT, which correlated with the higher levels of the SA accumulation and the elevated expression of the corresponding PR genes upon infection with the mutant virus.

### 3.5. Suppression of the SA-Modulated Host Response to CTV Infection is Mediated at the Level of the LMT1 RNA

To investigate whether the LMT1 RNA plays a role in the negative regulation of the SA-dependent host defense, we overexpressed LMT1 by agroinfiltrating *N. benthamiana* leaves using a binary construct harboring a cDNA of the LMT1 sequence placed under the CaMV 35S promoter (pLMT1; Figure 5A). The LMT1 sgRNA is not known to produce any protein. Although its nucleotide sequence overlaps with the beginning of the ORF 1a that encodes a 349 kDa-polyprotein, *in silico* translation of this RNA shows no ORFs in any of the three potential reading frames (data not shown). Nevertheless, to confirm that the phenomenon observed above is mediated at the RNA level, we generated two constructs that expressed mutant versions of LMT1 carrying substitutions within the original start codon for ORF 1a (nt 108–110; pLMT1M1) or the original start codon and the next potential start codon (nt 108–110 and 264–266; pLMT1M2) (Figure 5A). To verify expression of the LMT1 mutants, total RNA extracted from agroinfiltrated *N. benthamiana* leaves was subjected to RT-PCR. Robust amplification of the corresponding fragments was observed from the wild-type LMT1 and the two mutant constructs (Figure 5B). Next, we compared the levels of SA and mRNAs of the PR genes in *N. benthamiana* plants infiltrated with the *Agrobacterium* bearing pLMT1 or its mutant versions. Since *Agrobacterium* by itself was shown to induce SA and, consequently, the expression of SA-responsive genes [55], the *Agrobacterium* transformed with an “empty” binary vector plasmid was used as a control. As shown in Figure 5C,D, expression of LMT1 resulted in down-regulation of the transcripts of the three PR genes and a drastic reduction of the SA level, compared to that in the leaves infiltrated with the “empty vector” (EV), supporting our hypothesis that the inhibition of the SA accumulation and the expression of these host genes by the wild-type CTV was due to the production of LMT1. Importantly, both LMT1M1 and LMT1M2 also suppressed up-regulation of the three PR genes, and there was no significant difference in their levels in the leaves treated with each LMT1 variant (Figure 5C), suggesting that the suppression was mediated by the LMT1 RNA. We also examined whether the expression of the LMT1 RNA would have an effect on the infectivity of a secondary pathogen similar to that found with CTV-WT. *N. benthamiana* plants were infiltrated with the *Agrobacterium* cultures harboring an empty vector plasmid or pLMT1 followed by subsequent infiltration of *P. syringae* into the same leaves at four dpi. At three days after the subsequent infiltration, the leaves expressing LMT1 demonstrated more severe symptoms of *P. syringae* infection (Figure 5E) and a significantly higher CFU count than that in the leaves expressing the empty vector-transformed *Agrobacterium* (Figure 5F). Taken together, these results suggested that the CTV LMT1-triggerred modulation of host immunity occurs at the RNA level via suppression of the SA-signaling pathway.

### 3.6. LMT1 Modulates the Expression of Alternative Oxidase (AOX-1a) Gene and Production of Reactive Oxygen Species

As shown previously, SA can promote plant resistance to viruses (reviewed in [12]). SA inhibits the respiratory electron transport chain [63,64] resulting in the increase in reactive oxygen species (ROS) which subsequent signaling is proposed to trigger the induction of nuclear genes that affect viral replication and movement [17]. Based on this, we hypothesized that the higher SA level in the leaves infiltrated with the virus lacking LMT1 could have resulted in a more severe ROS accumulation compared to that in leaves infected by CTV-WT. To confirm this, the leaves infiltrated with CTV-WT and CTV-LMT1d were stained with DAB, which detects hydrogen peroxide, the most abundant ROS (Figure 6). For this test, the infiltrated leaves were collected at 7 dpi, a time point in the early infection stage at which the difference between the titers of the two viruses was not statistically significant (Figure 2E), yet LMT1 had already accumulated in a large amount in the wild-type virus-infected cells (Figure 1C). As a result, the areas infiltrated with CTV-LMT1d showed the development of dark brown staining compared to the light yellow-brown color in the areas infiltrated with CTV-WT (Figure 6A). The intensity of the DAB staining measured over a single line plot (Figure 6A; black dotted line) demonstrated higher values of brown pigmentation in the CTV-LMT1d-infiltrated region (Figure 6B). These results indicated that during the establishment of the CTV-LMT1d infection ROS burst was much more extensive compared to that upon the development of the CTV-WT infection.

Besides the SA signaling, ROS concentration in the mitochondrion is regulated by another critical enzyme—an alternative oxidase (AOX). AOX was reported to function as a suppressor of ROS accumulation upon virus infection [18,65]. Here, we analyzed the mRNA levels of the *AOX-1a* gene in the leaves infiltrated by the two virus variants as well as the constructs expressing the LMT1 RNA and its mutants. Remarkably, the expression level of *AOX-1a* in the leaves infected by CTV-WT was significantly higher compared to that in the CTV-LMT1d-infected leaves (Figure 6C). Furthermore, the increase in the *AOX-1a* expression level was also found in the leaves infiltrated with pLMT1 and its mutant versions (Figure 6D). Altogether, these results demonstrated that LMT1 stimulated the increased expression of AOX-1a and suppressed the ROS accumulation.

## 4. Discussion

### 4.1. Role of lncRNAs in Virus–Host Interactions

Compared to plant and animal genomes, those of viruses have significantly smaller sizes. For this reason, most if not all virus-encoded factors such as proteins or RNA sequences are justified. Viral proteins are often multifunctional and engaged in many different processes at different stages of the virus infection cycle. The same could be true for viral lncRNAs that represent an exciting yet not well understood area. One of the crucial tasks of a virus is to combat a defense response that the host mounts upon pathogen perception. To this end, in addition to a long list of viral proteins shown to counter-act host immune receptor signaling or antiviral RNA silencing, many of the studied viral lncRNAs appear to be active participants of the virus–host arms race as well. Such RNAs, perhaps less immunogenic, could carry or form a number of secondary/tertiary structures that bind cellular proteins, thus, effectively modulating their involvement in the cellular processes, including antiviral defense. For instance, adenovirus lncRNA VAI binds to the host protein kinase R, which is known to block cellular translation in response to virus infection through phosphorylation of eIF2, and inhibits its activity, thereby allowing viral protein synthesis [66,67]. Furthermore, flavivirus sfRNA decreases the antiviral activity of type I interferon by antagonizing a group of cell proteins, including dampening of the retinoid acid-inducible gene I-mediated interferon induction [68,69,70]. In addition, sfRNA, which possesses a complex secondary structure, serves as a decoy for the RNA silencing machinery and, thus, diverts the latter from targeting viral genomes in the insects transmitting the virus [71]. A similar mode of action has been also proposed for the ~ 600 nt 8S RNA of CaMV [8,72,73].

### 4.2. LMT1 Plays a Role in the Accumulation and Movement of CTV in Plant Hosts Through Inhibition of SA Signaling

Our study shows that LMT1, a lncRNA of CTV, facilitates virus infection by some means, which include modulation of the host immune response. LMT1 greatly increases invasiveness of CTV in the herbaceous host and is absolutely required for its infection of the natural citrus host. Although the LMT1-deficient virus was able to spread in *N. benthamiana*, there was an impediment in the establishment of systemic infection, compared to that by the wild-type CTV. This correlated with less efficient cell-to-cell movement and low occurrence of the mutant virus variant in cells forming the vascular bundles, suggesting that hindered ingress of CTV-LMT1d into the vascular system in the initially inoculated leaves affected the development of systemic infection. Importantly, virion formation was not altered in the mutant virus, and, thus, such virus property cannot be attributed for the phenotype observed. On the other hand, apparent deficiency in the accumulation and movement of the LMT1-deficient virus was accompanied by the elevated levels of SA and the SA-responsive genes, which were significantly higher than those upon the wild-type infection, implicating LMT1 in modulation of SA signaling. The results of further experiments using transient expression of the LMT1 RNA proved its ability to suppress the accumulation of this hormone compound and pointed to the role of LMT1 in facilitating virus accumulation and movement by influencing the host-virus interplay.

The observations made in this work are in agreement with previous studies, which examined incompatible as well as compatible plant host-virus systems and demonstrated that induction of phytohormonal signaling could inhibit virus accumulation and/or movement and hamper virus host invasion [14,16,18,22,74,75]. Moreover, for tobacco mosaic virus (TMV) in tobacco, cucumber mosaic virus (CMV) in squash, and potato virus Y (PVY) in potato, virus accumulation in the inoculated leaves of a susceptible host was shown to be reduced upon the application of exogenous SA [18,76,77]. Treatment with SA also had a profound effect on the cell-to-cell movement of TMV in *N. benthamiana* and tomato ringspot virus in tobacco and hindered the systemic translocation of TMV in *N. benthamiana*, CMV in tobacco and squash, and potato virus X (PVX) in tomato [14,15,16,18,75,78]. On the other hand, interfering with the SA pathways resulted in lowering the expression of corresponding PR genes and the increase of virus accumulation and systemic spread in the plum pox virus-*n. tabacum*, CMV-, turnip crinkle virus- or turnip mosaic virus-*Arabidopsis*, PVY-potato, and bamboo mosaic virus- or TMV-*N. benthamiana* pathosystems [74,76,79,80].

### 4.3. LMT1 Facilitates Virus Infection through the Induction of AOX-1a and Mitigation of Reactive Oxygen Species (ROS) Accumulation

As we demonstrated recently, infection by CTV triggers a signaling cascade resulting in ROS accumulation, and three viral proteins—p33, p23, and p20—contribute to this reaction [47]. Here, we show that prevention of the LMT1 production escalates the host immune response beyond that upon infection with the wild-type virus, and that LMT1 functions as an antagonist of the SA-mediated signaling. Interestingly, there are few other examples of viral factors that regulate plant innate immunity to promote virus infection. One of them is the p6 protein of CaMV, which was found to reduce the accumulation of SA as well as the expression of SA-dependent genes and the programmed cell death [22]. Another example is the CMV 2b protein shown to significantly reduce the inhibitory effect of SA on virus accumulation [21]. Our study, however, provides the first example implicating a plant virus lncRNA in the interference with the antiviral phytohormone-mediated defense.

Induction of the PR genes expression observed in this work represents a hallmark of the SA-mediated defense response in plants against many plant pathogens. However, the antiviral activity of the PR proteins remains elusive. On the other hand, another branch of the SA signaling that involves AOX and ROS were shown to have a role in the interactions between a number of plant viruses and their hosts [21,74,81,82,83,84,85]. Upon virus invasion, AOX can suppress programmed cell death by limiting production of ROS and, thus, promote virus infection as it was demonstrated with TMV in leafy mustard [85] and PVX and TMV in *N. benthamiana* [82,84,86]. In our study, expression of the LMT1 RNA by the wild-type virus resulted in the increase in the *AOX-1a* levels, which was accompanied by lowering of ROS accumulation. We, therefore, hypothesize that this chain of events promoted virus movement from the initially infected cells to the neighboring cells and further into the vascular system allowing the virus to effectively spread throughout the plant. The proposed model of the effect of LMT1 on the plant innate immunity, which takes advantage of the established innate immunity pathways involving SA, AOX, and ROS (reviewed in [17,87]), is presented in Figure 7. Production of LMT1 at the early stage of the CTV-WT infection suppresses the accumulation of SA, which negatively affects the extent of the downstream immune responses, including the accumulation of ROS. Additionally, LMT1 elevates the expression of AOX, which, in its turn, also lessens ROS burst. Thus, the expression of LMT1 mitigates the expression of antiviral defense genes and supports the wild-type virus accumulation and spread in the hosts. In contrast, the inability of the LMT1-lacking virus to counter-act or prevent early ROS accumulation escalates the plant immune responses resulting in a decreased efficiency of its infection in *N. benthamiana* and complete block of citrus infection.

## 5. Conclusions

Our study examined the role of LMT1, a lncRNA of one of the largest RNA viruses in its pathogenicity. Although LMT1 was discovered more than two decades ago, its functional importance was not understood, and for a long time this RNA has been considered as a by-product of a complex transcription strategy in a closterovirus. Here, we show that LMT1 functions in modulation of the host immune response to promote virus infection. Specifically, LMT1 antagonizes the SA-mediated signaling. To the best of our knowledge, CTV LMT1 is the first example of a viral lncRNA implicated in regulation of this branch of the plant innate immunity. We hypothesize that LMT1 has evolved as a consequence of virus adaptation to a perennial woody host with complex innate system barriers.

## Figures and Tables

**Figure 1 viruses-11-00436-f001:**
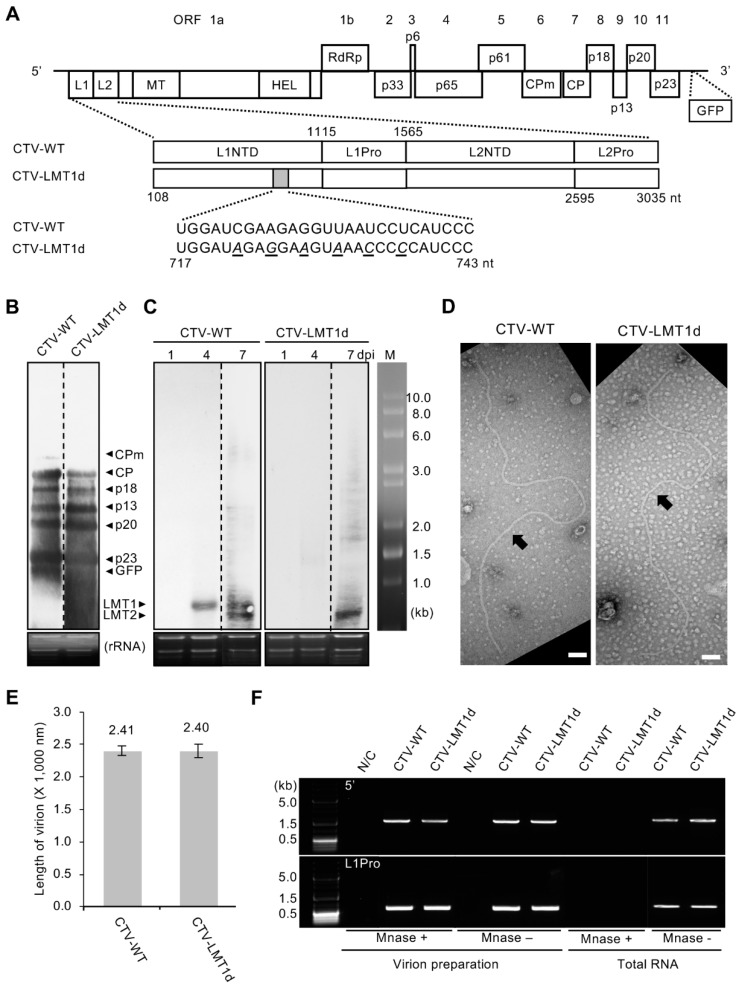
Prevention of LMT1 production does not affect virus ability to replicate or assemble proper virions. (**A**) Schematic representation of pCTV-WT and pCTV-LMT1d. The boxes represent reading frames (ORFs) and their translated products. L1 and L2, papain-like leader proteases; NTD, *n*-terminal domain; Pro, C-terminal proteolytic domain; MT, methyltransferase; HEL, helicase; RdRp, RNA-dependent RNA polymerase; p65, HSP70 homolog; CPm, minor coat protein; CP, coat protein. The silent nt mutations (shown as underlined italicized letters) introduced to modify the controller element of LMT1 (grey box) are shown in detail under the enlarged genome segment. All viruses have a green fluorescent protein (GFP) ORF inserted between the p23 gene and the 3′ non-translated region of the *Citrus tristeza virus* (CTV) genome. (**B**,**C**) Northern blot analysis of the total RNA isolated from *N. benthamiana* plants expressing CTV-WT and CTV-LMT1d hybridized with the 3′ positive-strand RNA-specific probe (**B**) or 5′ positive-strand RNA-specific probe (**C**). Total RNA used in (**B**) was extracted from samples collected at 7 dpi. sgRNAs for the corresponding ORFs are indicated. Lanes from different membranes are divided with dotted lines. (**D**) Examination of the virions produced by CTV-WT and CTV-LMT1d using negative contrast electron microscopy. Virions were purified from *N. benthamiana* leaves as described in Materials and Methods and examined using transmission electron microscopy (TEM). Representative images are shown. Virions are denoted by black arrows. Bars = 100 nm. (**E**) A mean length of virions calculated based on the TEM images of ten longest virions per treatment +/− standard error of the mean (*n* = 10). (**F**) Viral genomic RNA protection analysis using ribonuclease (MNase, NEB) treatment of purified virions followed by RT-PCR. Panel ‘5′’ shows the amplification using a pair of oligomers amplifying the 5′ terminal region of the CTV genome. Panel ‘L1Pro’ shows the amplification using a pair of oligomers amplifying a region of the genome encoding the protease domain of the L1 protease (Figure 1A; L1Pro). A “no-template” control is indicated as N/C.

**Figure 2 viruses-11-00436-f002:**
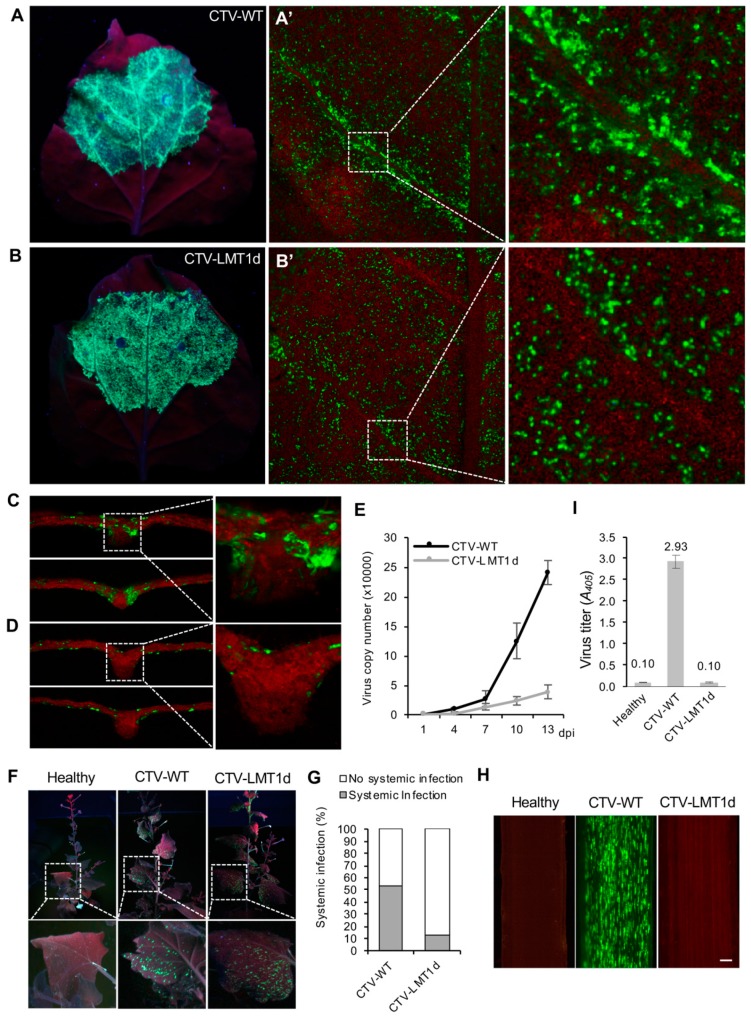
The LMT1-deficient virus shows an impediment in its invasiveness. (**A**–**D**) Expression of GFP in the *N. benthamiana* leaves infiltrated with the GFP-tagged CTV-WT or CTV-LMT1d. Images of GFP fluorescence were taken using a hand-held ultraviolet (UV) lamp (**A** and **B**) or fluorescence microscope (**A’**, **B’**, **C**, **D**) from the abaxial leaf surface (**A**, **A’**, **B**, **B’**) or a cross-section of a lateral leaf vein (**C**,**D**). (**E**) Accumulation of viruses in *N. benthamiana* plants [viral load = number of genomic RNA copies per ng of total RNA]. Error bar represents standard error of the mean (*n* = 3) within a set of samples consisting of one leaf disc per plant from three plants infiltrated with each virus variant. The absolute number of CTV RNA copies was determined by reverse transcription quantitative polymerase chain reaction (RT-qPCR) based on a standard curve generated by amplification of known amounts of the target under conditions identical to those of the experimental sample as described in Materials and Methods section. For each set of repeats, three RT-qPCR determinations were made. Three repeat experiments showed similar results. (**F**) Observation of systemic infection by the GFP-tagged CTV variants in *N. benthamiana*. Images show GFP fluorescence observed in the un-inoculated upper leaves of *N. benthamiana* plants compared to the healthy control plant (Healthy) showing no fluorescence. (**G**) A percentage of plants that developed systemic infection upon inoculation with CTV-WT or CTV-LMT1d calculated at 16 dpi. A total of 60 plants per each virus were tested in three independent agroinfiltrations. (**H**) Images show GFP fluorescence observed in the phloem-associated cells of virus-inoculated *C. macrophylla* plants at 12 months after inoculation. Image on the left represents a control mock-inoculated plant (Healthy). Observations were made on the internal surface of bark using a dissecting fluorescence microscope. Bars = 0.3 mm. (I) Analysis of virus titer in plants shown in (**H**) using enzyme-linked immunosorbent assay (ELISA) with the CTV-specific antibody. Each bar represents a mean of the virus titer (*A*_405_ values of ELISA) +/− standard error of the mean of samples obtained from all biological replicates, *n* = 28 for CTV-WT, *n* = 10 for un-inoculated healthy plants, *n* = 106 for CTV-LMT1d.

**Figure 3 viruses-11-00436-f003:**
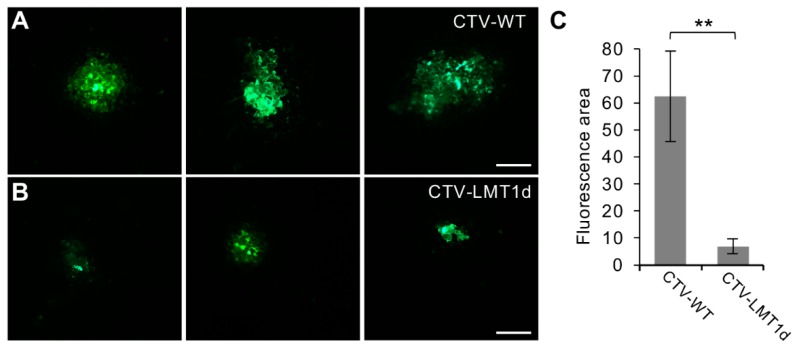
Assessment of CTV-WT (**A**) and CTV-LMT1d (**B**) cell-to-cell movement in the *N. benthamiana* leaves. Images show three representative GFP-expressing foci observed at 21 days after rub-inoculation of equal-concentration virion suspensions of CTV-WT or CTV-LMT1d in the mesophyll cells of *N. benthamiana* plants using fluorescence microscope. Bars = 0.1 mm. (**C**) Quantitative measurement of GFP fluorescence in the images shown in (**A**) and (**B**). Each bar represents a mean value +/− standard error of the mean calculated from all images of observed GFP foci (*n* = 59 and 32 for CTV-WT and CTV-LMT1d, respectively). Statically significant differences determined by Student’s t-test, *p* < 0.01, were denoted by asterisk (**).

**Figure 4 viruses-11-00436-f004:**
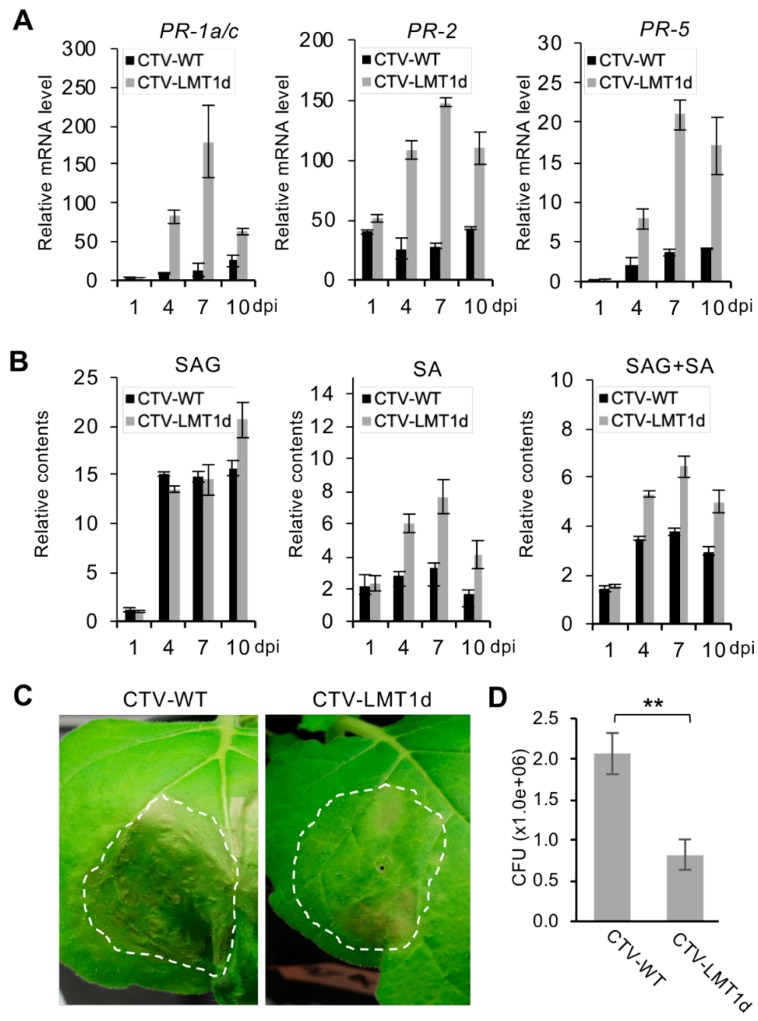
Comparison of SA-mediated defense response in the *N. benthamiana* leaves infiltrated with pCTV-WT and pCTV-LMT1d at 1, 4, 7 and 10 dpi. (**A**) Relative mRNA levels of *PR-1a/c*, *PR-2*, and *PR-5* genes. (**B**) Relative contents of salicylic acid (SA) and glucose-conjugated SA (SAG). Black bar, pCTV-WT; grey bar, pCTV-LMT1d. Each bar represents a mean value +/− standard error of the mean (*n* = 3) within a set of samples consisting of one leaf disc per plant from three plants infiltrated with each virus variant. For each repeat, three RT-qPCR determinations were made. Three repeat experiments showed similar results. (**C**) Symptoms of *P. syringae* in *N. benthamiana* plants. Plants pre-infiltrated with a virus construct were challenged with *P. syringae* at 7 dpi and assayed at 4 days after the challenge. (**D**) Titers (colony-forming units (CFU) per leaf disc) of *P. syringae* in the leaves shown in (**C**). Each bar represents a mean value +/− standard error of the mean (*n* = 9) within a set of samples consisting of one leaf disc per plant from three plants infiltrated with each treatment collected from three independent replicates. Statistically significant differences determined by Student’s *t*-test, *p* < 0.01, were denoted by asterisk (**).

**Figure 5 viruses-11-00436-f005:**
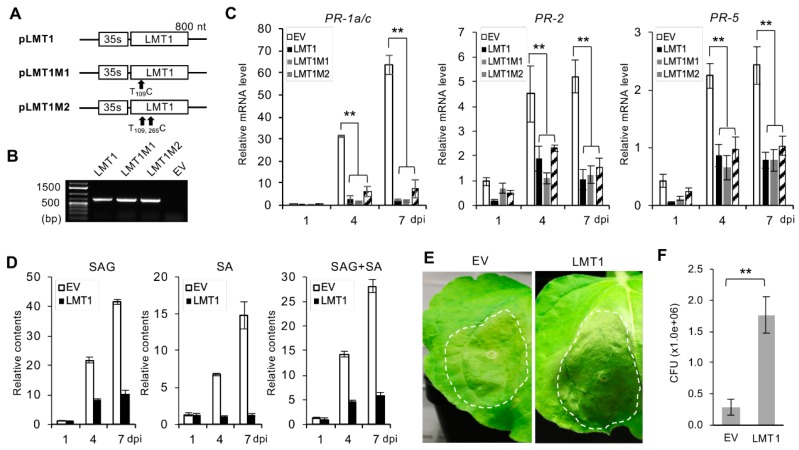
Suppression of SA-mediated defense response by the LMT1 RNA. (**A**) Schemes of the expression constructs of LMT1 (pLMT1) and its mutants (pLMT1M1 and pLMT1M2). (**B**) RT-PCR analysis of the LMT1 transcripts. LMT1-specific primers were used to amplify the transcripts from the total RNA extracted from *N. benthamiana* plants infiltrated with the constructs shown in (**A**) along with an empty vector. The size of the expected amplification product was 700 bp. (**C**,**D**) Comparison of the relative mRNA levels of *PR-1a/c*, *PR-2* and *PR-5* genes (**C**) and relative contents of salicylic acid (SA) and glucose-conjugated SA (SAG) (**D**) upon infiltration of *N. benthamiana* with pLMT1 and its mutants at 1, 4, and 7 dpi. Open bar, “empty-vector” *Agrobacterium* control (EV); black bar, pLMT1; grey bar, pLMT1M1; bar filled with diagonal stripes, pLMT1M2. Statically significant differences determined by Student’s t-test, *p* < 0.01, were denoted by asterisks (**). (**E**) Symptoms of *P. syringae* infection in *N. benthamiana.* Plants pre-infiltrated with a LMT1 construct or empty vector were challenged with *P. syringae* at 4 dpi and assayed at 3 days after the challenge. (**F**) Titers (CFU per leaf disc) of *P. syringae* in the leaves shown in (**E**). Sample analysis was done as described in Figure 4.

**Figure 6 viruses-11-00436-f006:**
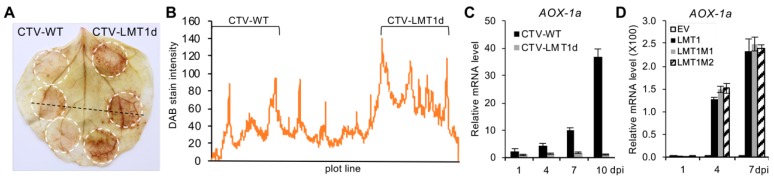
The effect of LMT1 on the accumulation of reactive oxygen species (ROS) and the expression of alternative oxidase (*AOX-1a*) gene. (**A**) ROS accumulation was demonstrated using 3,3′-diaminobenzidine (DAB) staining of the leaves infiltrated with CTV-WT or CTV-LMT1d. A representative image taken at 7 dpi is shown. The left and the right halves of the leaf were infiltrated with CTV-WT and CTV-LMT1d, respectively. A dotted black line is the plot was used for the quantification of staining intensity. (**B**) Relative DAB staining intensity measured from the image shown in (**A**). The value was determined by measuring the pixel intensity along the line plot. The infiltration spots are indicated by the brackets above the graph. (**C**) Relative mRNA levels of *AOX-1a* gene in the leaves infiltrated with CTV-WT and CTV-LMT1d at 1, 4, 7 and 10 dpi. Black bar, pCTV-WT; grey bar, pCTV-LMT1d. (**D**) Relative mRNA levels of *AOX-1a* gene in the leaves infiltrated with pLMT1 and its mutants at 1, 4 and 7 dpi. Open bar, “empty-vector” *Agrobacterium* control (EV); black bar, pLMT1; grey bar, pLMT1M1; bar filled with diagonal stripes, pLMT1M2. Each bar represents a mean value +/− standard error of the mean (*n* = 3) within a set of samples consisting of one leaf disc per plant from three plants infiltrated with each treatment. For each set of repeats, three RT-qPCR determinations were made. Three repeat experiments showed similar results.

**Figure 7 viruses-11-00436-f007:**
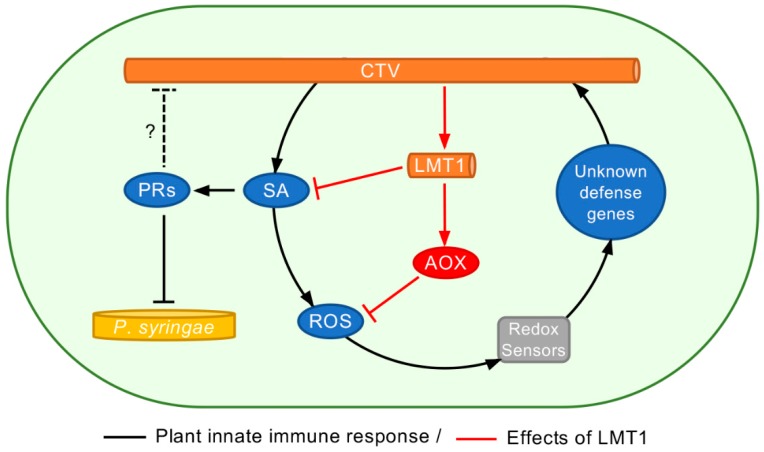
Schematic model of the CTV interplay with the plant innate immunity. CTV invasion triggers innate immune responses initiated by SA, which acts against virus accumulation and movement. Early production of LMT1 inhibits the accumulation of salicylic acid and also enhances the expression of an alternative oxidase, which suppresses the accumulation of reactive oxygen species. Plant innate immune responses are indicated in black lines. The effects by LMT1 are indicated in red lines. A possible effect of the PR proteins on the CTV infection is indicated as a black dotted line. PR, pathogenesis-related proteins; SA, salicylic acid; AOX, alternative oxidase; ROS, reactive oxygen species.

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
