# Peer review of "A Long Non-Coding RNA of Citrus tristeza virus: Role in the Virus Interplay with the Host Immunity"

_viruses, 2019, doi:10.3390/v11050436_

Reviewer 1 Report

The manuscript viruses-490114 describes the role that the low-molecular-weight tristeza 1 (LMT1) non coding subgenomic RNA from CTV isolate T36 plays in the host immunity of N. benthamiana and Citrus macrophyllla. LMT1 functions in the modulation of the host immune response to promote virus infection by antagonizing the SA-mediated signaling. The results presented are sound, novel and the manuscript is written in a very structured and clear manner. 

Author Response

We thank the reviewer for the positive comments about our manuscript.

Reviewer 2 Report

In this article, Kang and collaborators studied the biological role of a viral long non-coding RNA, termed LMT1, expressed by CTV. For their analyses they followed two classical, but effective, approaches: (i) knocking out the expression of LMT1 in the context of a CTV infection, and (ii) the 35S-driven over expression of LMT1. Interestingly, results from these two approaches point in the same direction: LMT1 plays a role in counteracting the activation of SA-mediated defense signaling during CTV infection. Based on their results, authors propose that LMT1 is acting at two different levels: (i) preventing the accumulation of SA and (ii) inducing the expression of AOX for the further limitation of ROS production.

First of all I have to say that the article is well written, well organized, and easy to follow. On the other hand, I believe that authors need to carry out a crucial experiment to wrap this article up: infection of Nicotiana benthamiana plants deficient in SA (for instance, NahG plants). If the hypothesis is true, the expected results here would be that CTV-LTM1d is not longer attenuated and recover its capacity to infect this SA-defective plants as well as the wild type CTV. If this experiment can be done and included, I have no doubt that this article deserves to be published in "Viruses".

Other comments:

- Line 14-15. Authors state that CTV produces a unique lncRNA, LMT1. However there is another one, LMT2, as mentioned latter in the paper. Please, correct this.

. Line 415-445. From the experiments explained here, authors suggest that LMT1 plays a role in facilitating CTV cell-to-cell movement. It is a bit confusing the way that these conclusions are presented, since a real implication in cell-to-cell movement points rather to a role of LMT1 in something related with, for instance, the proper movement of the virus through plasmodesmata. The observed defect might have nothing to do with movement but it might be related with replication/translation and/or suppression of a host antiviral mechanism (as finally suggested by further results) . Please, clarify this for a better understanding.

- Figure 4: There is a missing control in this figure. Authors need to include the empty vector in parallel for proper comparison of the effect of CTV-WT and CTV-LMT1 over the different variables (PRs, SA, SAG, etc). It is not clear whether the observed induction are produced by the CTV infection or by the agrobacteria it self.

- Figure 5: This reviewer thinks that it would be elegant and informative to express the RNA that corresponds to the reverse complementary sequence of LMT1, which is supposed to be not expressed during CTV infection. Another interesting RNA to be expressed and tested is the one for LMT2.

Author Response

Reviewer #2: In this article, Kang and collaborators studied the biological role of a viral long non-coding RNA, termed LMT1, expressed by CTV. For their analyses they followed two classical, but effective, approaches: (i) knocking out the expression of LMT1 in the context of a CTV infection, and (ii) the 35S-driven over expression of LMT1. Interestingly, results from these two approaches point in the same direction: LMT1 plays a role in counteracting the activation of SA-mediated defense signaling during CTV infection. Based on their results, authors propose that LMT1 is acting at two different levels: (i) preventing the accumulation of SA and (ii) inducing the expression of AOX for the further limitation of ROS production.

First of all I have to say that the article is well written, well organized, and easy to follow. On the other hand, I believe that authors need to carry out a crucial experiment to wrap this article up: infection of Nicotiana benthamiana plants deficient in SA (for instance, NahG plants). If the hypothesis is true, the expected results here would be that CTV-LTM1d is not longer attenuated and recover its capacity to infect this SA-defective plants as well as the wild type CTV. If this experiment can be done and included, I have no doubt that this article deserves to be published in "Viruses".

We thank the reviewer for the comments. In our study, we demonstrated that LMT1 counter-acts the plant immune response to CTV by two different means. One of them is suppression of the SA accumulation, and the other is mitigation of ROS accumulation via the induction of AOX-1a (Figure 7). From our results shown in Figure 6, we proposed that lowering ROS accumulation promoted virus systemic infection by facilitating virus translocation to the vascular system. Because SA is not the only determining factor for the ROS level, if the proposed experiment with SA-deficient Nicotiana benthamiana plants (NahG plants) could not result in the full restoration of CTV-LMT1d infectivity to the level of that of CTV-WT, the result cannot refute our conclusion as, in addition to lowering the SA level, activation of AOX-1a branch will likely be needed.

Other comments:

- Line 14-15. Authors state that CTV produces a unique lncRNA, LMT1. However there is another one, LMT2, as mentioned latter in the paper. Please, correct this.

We changed the text as suggested by removing ‘unique’.

Line 415-445. From the experiments explained here, authors suggest that LMT1 plays a role in facilitating CTV cell-to-cell movement. It is a bit confusing the way that these conclusions are presented, since a real implication in cell-to-cell movement points rather to a role of LMT1 in something related with, for instance, the proper movement of the virus through plasmodesmata. The observed defect might have nothing to do with movement but it might be related with replication/translation and/or suppression of a host antiviral mechanism (as finally suggested by further results). Please, clarify this for a better understanding.

We thank the reviewer for this comment and understand the source of such confusion. From a stand point of the classical definition originated with the studies on the 30K protein of Tobacco mosaic virus, viral movement factors, which in most cases are represented by viral proteins, interact with plasmodesmata to mediate virus cell-to-cell transport. However, as it was shown in numerous studies, besides the need for assistance to pass through plasmodesmata, virus spread or movement often depends on the ability of viral factors to counteract plant defense mechanisms. Thus, it was suggested that viral factors involved in virus movement can act through a variety of viral/cellular functions, which leads to a broader definition of viral movement factors. 

- Figure 4: There is a missing control in this figure. Authors need to include the empty vector in parallel for proper comparison of the effect of CTV-WT and CTV-LMT1 over the different variables (PRs, SA, SAG, etc). It is not clear whether the observed induction are produced by the CTV infection or by the agrobacteria itself.

It is known that Agrobacterium itself induces expression of the PR genes and elevates SA/SAG (also shown in Figure 5 of our study). The CTV-WT and CTV-LMT1d viruses were introduced into plants using the same O.D. levels of Agrobacterium. Therefore, the effect of the Agrobacterium on the PR genes and SA/SAG would be expected to be equal in both treatments. The bars in Figure 4 combine the effect of Agrobacterium plus the effect of the respective virus. As the effect of Agrobacterium is similar for both virus variants, such data presentation allows to point out the differences in the parameters being measured between the two viruses and the effect of a lack of LMT1 on those.

- Figure 5: This reviewer thinks that it would be elegant and informative to express the RNA that corresponds to the reverse complementary sequence of LMT1, which is supposed to be not expressed during CTV infection. Another interesting RNA to be expressed and tested is the one for LMT2.

We thank the reviewer for this suggestion and will consider these additional experiments in our future research.

Round  2

Reviewer 2 Report

I would like to thanks the authors for their responses. I think that manuscript might be a little bit better if they do any of the proposed experiments, but in the end the manuscript also deserve publication as it is right now. Then, I would like this manuscript to be accepted for publication.